# Synthesis, In Silico and In Vivo Toxicity Assessment of Functionalized Pyridophenanthridinones via Sequential MW-Assisted Intramolecular Friedel-Crafts Alkylation and Direct C–H Arylation

**DOI:** 10.3390/molecules27238112

**Published:** 2022-11-22

**Authors:** Marlyn C. Ortiz Villamizar, Carlos E. Puerto Galvis, Silvia A. Pedraza Rodríguez, Fedor I. Zubkov, Vladimir V. Kouznetsov

**Affiliations:** 1Laboratorio de Química Orgánica y Biomolecular, CMN, Universidad Industrial de Santander, Parque Tecnológico Guatiguará, Km 2 Vía Refugio, Piedecuesta 681011, Colombia; 2Department of Organic Chemistry, Peoples’ Friendship University of Russia (RUDN University), 6 Miklukho-Maklaya Street, 117198 Moscow, Russia

**Keywords:** pyrido[3,2,1-*de*]phenanthridin-6-ones, *N*-aryl-*N*-(2-bromobenzyl) cinnamamides, intramolecular Friedel-Crafts alkylation, catalyzed direct C–H arylation, in silico computational methods, zebrafish embryos toxicity

## Abstract

A rapid, efficient, and original synthesis of novel pyrido[3,2,1-*de*]phenanthridin-6-ones is reported. First, the key cinnamamide intermediates **8a**–**f** were easily prepared from commercial substituted anilines, cinnamic acid, and 2-bromobenzylbromide in a tandem amidation and *N*-alkylation protocol. Then, these *N*-aryl-*N*-(2-bromobenzyl) cinnamamides **8a**–**f** were subjected to a TFA-mediated intramolecular Friedel-Crafts alkylation followed by a Pd-catalyzed direct C–H arylation to obtain a series of potentially bioactive 4-phenyl-4,5-dihydro-6*H*,8*H*-pyrido[3,2,1-*de*]phenanthridin-6-one derivatives **4a**–**f** in good yields. Finally, the toxicological profile of the prepared final compounds, including their corresponding intermediates, was explored through in silico computational methods, while the acute toxicity toward zebrafish embryos (96 hpf-LC_50_, 50% lethal concentration) was also determined in the present study.

## 1. Introduction

The exploration and study of the synthesis of nitrogen-containing polycyclic compounds is one of the major challenging objectives in current organic chemistry because of the interesting biological properties of these scaffolds [1]. For this purpose, simple and relevant *N*-heterocycles fused into one skeleton could serve as a useful platform for the discovery of new bioactive compounds and thereby design novel synthetic strategies for their preparation, an important and actual task [2,3,4]. In this sense, representative examples of the isoquinoline core, which is the main unit of aporphinoid alkaloids **1** [5,6] and the phenanthridine skeleton present in the structure of lycorines **2**, a group of amaryllidaceae alkaloids [7,8], have led to the discovery of a promising antitumor, antibacterial, and antiprotozoal agent (Figure 1).

Under this approach, our research group envisioned that the combination between quinolines and isoquinolines would be a good strategy in the search for biologically active nitrogen-containing polycyclic compounds, but at the same time, we considered that the power of this tool will be enhanced if the proposed structures were oxidized [9,10]. Thus, unknown nitrogen-containing tetracyclic molecules with the ring ABCD system can be assembled from the fusion of isoquinolin-1(2*H*)-one and quinoline, giving the corresponding pyrido[3,2,1-*de*]phenanthridin-8-one **3**, while the respective pyrido[3,2,1-*de*]phenanthridin-6-one **4** results from the fusion of isoquinoline and quinolin-2(1*H*)-one skeletons (Figure 2).

The phenanthridinone core is a common structure present in a wide range of natural alkaloids with numerous biological and pharmacological properties [11]. Phenanthridinone analogs are key scaffolds with a potential activity against neurodegenerative disease [12], and they are also antitumoral [13] and anti-HIV [14], with an immunomodulatory activity [15]. To the best of our knowledge, the chemistry, synthetic approaches, and biological activities of pyridophenanthridinones **3**–**4** remain unexplored, and just a few reports regarding the synthesis of system **3** have been reported so far [16,17], as they are analogs of aporphine alkaloids [18]. In the case of derivatives **4**, any attempts for their synthesis have failed or have not been reported yet. In contrast, several synthetic methodologies for the construction of quinolin-2-one, phenanthridine, aporphine, and lycorine skeletons have been developed in the past decades through different strategies [19,20,21,22,23,24,25,26,27,28,29,30].

For instance, the synthesis of dihydroquinolones and dihydroisoquinolones has been previously carried out by the direct intramolecular arylation reaction of the C–H bond through the palladium catalysis [31,32]. Along these lines, the intramolecular Friedel-Crafts hydroarylation (for dihydroquinolinones and phenanthridines) [33,34] and the direct C–H arylation (for phenanthridines, aporphines, and lycorines) [35,36,37] are valuable approaches that can be used for the rational synthesis design of pyridophenanthridinone derivatives.

In continuation of our recent studies on the synthesis of nitrogen-containing scaffolds [38,39], we focused our efforts on proposing and executing a retrosynthetic strategy for the efficient synthesis of pyridophenanthridin-6-ones involving a sequential intramolecular Friedel-Crafts alkylation and a direct C–H arylation starting from easily prepared *N*-aryl-*N*-(2-bromobenzyl)cinnamamides intermediates. Herein, we report a novel approach to accessing these interesting architectures and their in silico and in vivo toxicological properties using the zebrafish embryo model in order to disclose their potential in drug discovery.

## 2. Results and Discussion

### 2.1. Chemistry

In order to overcome the lack of a synthetic route to access pyridophenanthridin-6-ones **4**, in this work we initially envisioned two possible retrosynthetic scenarios in which the disconnections on the ACBD skeleton **4** could be achieved by C–C bond formation strategies based on the Friedel-Crafts hydroarylation and the direct C–H arylation. However, the real challenge would be to establish in which order these disconnections will be performed. In the first approach, the intramolecular C–H arylation reaction will furnish rings ABC, followed by the intramolecular Friedel-Crafts hydroarylation to generate rings CD, while in a second approach these steps could be shifted between them and performed the Friedel-Crafts hydroarylation first, followed by the C–H arylation reaction (Figure 1).

Regardless of the synthetic approach, both strategies converge on the respective *N*-aryl-*N*-(2-bromobenzyl) cinnamamide **7** as a key intermediate, which can also be assessed by two different routes: first, through an amidation reaction between commercially available anilines and cinnamic acids to furnish *N*-aryl cinnamamide **8**, followed by the corresponding *N*-alkylation with 2-bromobenzyl bromide (approach A, Figure 1), or through the amidation between cinnamic acid and the *N*-(2-bromobenzyl) aniline **9** prepared from commercial starting materials (approach B, Figure 1).

During our initial experiments, we first examined the optimal protocol for the amidation reaction to yield cinnamanilide **7**. This process is usually performed from cinnamoyl chloride and the corresponding anilines [40] or through the direct amidation between cinnamic acid and aryl amines in the presence of metal catalysts and activating agents, or even assisted by microwave irradiation, due to the low nucleophilicity of anilines [41,42,43,44,45]. Although we recently demonstrated the robustness of tris-(2,2,2-trifluoroethyl) borate (B(OCH_2_CF_3_)_3_) as a catalyst for the direct coupling between cinnamic acid and primary amines [46], this protocol did not give the expected results, and then, looking for a suitable green, efficient procedure, we focused our efforts on exploring the efficient approaches used in peptide synthesis for coupling unreactive amines with cinnamic acids. Therefore, we selected 2-(1*H*-benzotriazole-1-yl)-1,1,3,3-tetramethylaminium tetrafluoroborate (TBTU) as a mild and effective coupling reagent for the direct condensation of cinnamic acid **10** with *N*-(2-bromobenzyl) anilines **9** and commercial anilines **11** under microwave (MW) reaction conditions [47] (Figure 2). Although *N*-substituted aniline **9** was prepared in excellent yield through the direct *N*-alkylation of aniline **11a** with 2-bromobenzyl bromide **12** promoted by cesium carbonate (Cs_2_CO_3_) [48], this intermediate resulted to be unreactive towards cinnamanilide **7**, probably due to the steric hindrance and low nucleophilic nature of **9** (route a, Figure 2).

In contrast, a series of *N*-aryl cinnamamides **8a**–**f** were obtained in excellent yields (83–98%) when the amidation reaction between cinnamic acid **10** and substituted anilines **11a**–**f** was assisted by MW irradiation in the presence of TBTU and Et_3_N in small quantities of DMF (See Appendix A). Then, the subsequent *N*-alkylation of amides **8a**–**f** with 2-bromobenzyl bromide **12** was successfully performed in the presence of potassium t*ert*-butoxide (*t*-BuOK) in THF for 4 h at 70 °C to furnish the desired *N*-aryl-*N*-(2-bromobenzyl) cinnamamides **7a**–**f** in excellent yields (route b, Figure 2).

With the *N,N*-disubstituted cinnamamides **7a**–**f** in hand, we had the required scaffold with all the features needed to build the targeted pyridophenanthridin-6-ones **4**. According to our retrosynthetic analysis (Figure 1), in the first approach we envisioned that the ABC ring system could be generated through the C–C bond formation by transition-metal catalyzed intramolecular cross-coupling reaction between rings A and C. Within this field, there are several elegant reports regarding the use of palladium catalytic systems, and among them, the intramolecular direct C–H bond arylation using Pd(OAc)_2_/P(Cy)_3_ (P(C₆H₁₁)₃) is one of the most recognized protocols [27,28,29,30]. In our previous report, we found that the complex PdCl_2_(MeCN)_2_ efficiently catalyzed the intramolecular C–H arylation reaction for the construction of the tricyclic benzo[*c*]chromene core from 2-bromoaryl-benzyl ethers [49]. Thus, we replicated these established reaction conditions to promote the intramolecular direct C–H arylation of *N*,*N*-disubstituted cinnamamide **7** and access to the corresponding 5,6-dihydrophenanthridine **5**. Unfortunately, this reaction did not proceed as expected, probably due to the presence of the alkene moiety that could also interact and deactivate the Pd-catalyst towards the C–H arylation (Figure 3).

Having to discard this approach to the access to pyridophenanthridin-6-ones **4**, we next focused our efforts on evaluating approach B by studying the intramolecular Friedel-Crafts hydroarylation of *N*,*N*-disubstituted cinnamamide **7** to generate the ring system CD in the first place. Generally, this strategy has been reported using both Brønsted acids, such as polyphosphoric acid (PPA), sulfuric acid (H_2_SO_4_), triflic (CF_3_SO_3_H) or trifluoroacetic (TFA), and Lewis acids (AlCl_3_, In(OTf)_3_, Zr(SO_4_)_2_, zeolite (H-USY) as catalysts [50,51] Although the major drawbacks of these protocols are the generation of hazardous and toxic by-products, in pursuit of developing an efficient and environmentally friendly protocol, we selected TFA as a catalyst for the Friedel-Crafts hydroarylation due to its high dielectric constant, water solubility, low boiling point, and strong acidity [52], besides of being an inexpensive and common reagent/solvent used previously in the transformation of cinnamic acid esters and cinnamanilides [33].

In the first place, we evaluated the intramolecular cyclization reaction of **7a** in TFA under conventional heating at 50 °C, finding that after 48 h the desired product **6a** was obtained in 75% yield. However, using microwave irradiation to assist this reaction, we were able to significantly reduce the reaction time to 40 min by heating the system to 140 °C without any evidence of substrate or product degradation, obtaining the 3,4-dihydroquinolin-2(1*H*)-one **6a** with similar yield (70%) to the previous experiment under conventional heating. With these optimized reaction conditions, the *N*-benzyl-4-phenyl-3,4-dihydroquinolin-2(1*H*)-ones **6a**–**f** were obtained with an efficient and clean procedure in good yields (Figure 4).

Next, having transformed the alkene moiety to avoid the deactivation of the Pd-catalyst during the C–H arylation, we proceeded to replicate the same reaction conditions described in Figure 3 to promote the intramolecular C*_sp_*_2_–C*_sp_*_2_ coupling reaction. To our delight, using PdCl_2_(MeCN)_2_ (5 mol%) as a catalyst, PivOH (30 mol%) as an additive, P(Cy)_3_ (10 mol%) as a ligand, and K_2_CO_3_ (3 equiv.) as a base in *N*,*N*-dimethylacetamide (DMA) as a solvent, we successfully obtained the corresponding pyrido[3,2,1-*de*]phenanthridin-6-ones **4a**–**f**, achieving the synthesis of the desired and new nitrogen-containing tetracyclic derivatives with the ring system ABCD in excellent yields (93–99%) (Figure 4).

The reaction mechanism of intramolecular direct arylation catalyzed by palladium has been extensively reviewed [53]. Our previous experience allowed us to find a catalytic cycle composed of 3 main steps (Figure 5) [49]. The first one consists of an oxidative addition of the Pd^(0)^ to the 3,4-dihydroquinolin-2(1*H*)-one **6a** to form **A** and the subsequent anion exchange by the formation in situ of potassium pivalate (II) that leads to **B**. The critical step in this reaction is the concerted metalation-deprotonation (CDC) transition state where the C–H activation proceeds through a proton abstraction pathway as it was reported by Echavarren and coworkers [54]. Finally, a reductive elimination takes place and the new bond C*_sp_*_2_–C*_sp_*_2_ is formed to give **4a**.

All of the synthesized compounds, including cinnamamides **7** and **8**, 3,4-dihydroquinolin-2(1*H*)-ones **6**, and the pyridophenanthridin-6-ones **4,** were obtained as stable crystalline substances with well-defined melting points (See Appendix A). Their structures were confirmed using different spectral (IR, HRMS, and NMR analysis: ^1^H and ^13^C NMR spectra, as well as two-dimensional experiments (COSY, HMQC, and HMBC).

### 2.2. Computational Studies

#### 2.2.1. In Silico Prediction of Physicochemical Properties

Lipinski’s rule of five concerns a set of molecular descriptors widely used in drug discovery to describe the drug-likeness, biological activity, and bioavailability of new compounds [55]. Those parameters are molecular weight (MW) ≤ 500 Da, lipophilicity assessed by the cLogP ≤ 5 (the logarithm of the partition coefficient between water and octanol), water solubility ≥ −5, number of hydrogen bond donors (HBD) and acceptors (HBA), ≤5 and ≤10, respectively, number of rotatable bonds (nRB) ≤ 10, and topological polar surface area (TPSA) ≤ 140 [56,57]. According to these predictions (Table 1), obtained using the online SwissADME database [58], it was observed that concerning the molecular weight, expressed as MW in g/mol, none of the synthesized compounds **4**–**8** exerted MW values greater than 500 g/mol. The lipophilicity parameter, expressed as LogP, indicated that *N*-aryl-*N*-(2-bromobenzyl)cinnamamides **7b** and **7c**, and the series of derivatives **7f** and 3,4-dihydroquinolin-2(1*H*)-one **6f**, substituted with the ethyl group, exhibited values over 5, being the derivatives that violated the established parameters of Lipinski’s rule of five. Meanwhile, the rest of the derivatives, including the targeted pyrido[3,2,1-*de*]phenanthridin-6-ones **4a**–**f**, met the requirements established by the rule and can be considered as promising candidates and lead structures having drug-likeness properties (Table 1).

#### 2.2.2. Bioavailability Radar

The bioavailability radar is another tool included in the online SwissADME database that graphs the behavior and the drug-likeness of the tested molecules [59]. These snapshots are presented as a hexagon where six physicochemical properties such as lipophilicity, size, polarity, solubility, flexibility, and saturation at each vertex are considered. These criteria are very important for analyzing the drug-likeness of a molecule and defining its pharmacological potential. Within these charts, the pink area represents the optimal range for each property: size: MW between 150 and 500 g/mol, lipophilicity represented by cLog*p* not higher than 5.0, and solubility represented by cLogS higher than −5. Polarity (TPSA) should exhibit values between 20 and 130, while saturation, expressed as the fraction of *sp*^3^ hybridized carbons, should be not less than 0.25 and flexibility not more than 9 rotatable bonds.

In order to analyze how all the chemical transformations performed to obtain pyridophenanthridin-6-ones **4**, starting from cinnamamides **8**, we changed and improved the physicochemical properties of each intermediate until reaching the desired derivatives; the bioavailability radar for cinnamamide **8f**, *N*-(2-bromobenzyl) cinnamamide **7f**, 3,4-dihydroquinolin-2(1*H*)-one **6f,** and pyridophenanthridin-6-one **4f** are shown in Figure 3. In general, all of the synthesized compounds exhibited the same behavior, where the α,β-unsaturation present in cinnamamides **7f** and **8f** affected their drug-likeness properties (Figure 3A,B).

However, after the intramolecular Friedel-Crafts hydroarylation, the ratio of *sp*^3^ hybridized carbons increased and the Lipinski parameters of 3,4-dihydroquinolin-2(1*H*)-one **6f** were slightly enhanced (Figure 3C). Finally, the physicochemical values of pyridophenanthridin-6-one **4f** fell closely in the pink area of its bioavailability radar, allowing us to consider the series of pyridophenanthridin-6-ones **4a**–**f** as potential drug candidates (Figure 3D).

#### 2.2.3. ADMET Study

The ADMET profile predicts the absorption, distribution, metabolism, excretion, and toxicity properties of a potential bioactive compound, important criteria in the drug discovery field. The pharmacokinetic properties for all the synthesized compounds **4**–**8** were theoretically calculated with the online application PreADME and are included in Table 1 [60]. Human intestinal absorption (HIA) is one of the most important ADME properties in drug discovery since this parameter is related to how a potential drug could be transported through the intestines and represents an alternative indicator for oral bioavailability with an important role in preclinical trials [61].

Although there could be some differences between the predicted and experimental values, acceptable models have grouped the HIA values as 0–20% (poor), 20–70% (moderately), and 70–100% (good). In our study, all the tested substances can be classified as compounds with promising absorption properties since the HIA values exhibited are between 97–100% (Table 1). The next parameter analyzed was distribution. In a pursuit that the synthesized compounds can reach their biological target, so they can act and be metabolized, these xenobiotic substances need to cross from the bloodstream into the tissues by not getting bound to the plasma proteins.

Although the plasma protein binding (PPB) of exogenous compounds depends on the concentration of available binding proteins, the affinity constant of the drug for the proteins, the number of available binding sites, and the presence of pathophysiologic conditions or endogenous compounds may alter drug–protein interaction. These are parameters considered in computational calculations [62], so the degree of PPB was predicted for all the synthesized compounds based on these computational models, noticing that almost all of them exhibited PPB values above 90%, which indicates strong binding interactions with the plasma protein (not desired). Among the series of prepared compounds, (*E*)-3-(3,4-dimethoxyphenyl)-*N*-phenyl acrylamide **8e** showed a weak plasma protein interaction (PPB = 81.27%), suggesting its facile diffusion and easy transport to its corresponding biological target (Table 1). Another relevant factor in drug design is the ability of small molecules to permeate the Blood Brain Barrier (BBB), since compounds that can pass across the vessels that vascularize the central nervous system (CNS) can be classified as CNS-active agents. The transport of molecules through the BBB is coordinated by the endothelial cells (ECs), which regulate the CNS homeostasis, and this restrictive property of the BBB results in an obstacle to drug delivery to the CNS, especially during the treatment of brain tumors [63]. The complexity of the BBB complicates CNS drug delivery, and major efforts have been made to generate methods to modulate or bypass the BBB for the delivery of small molecules. With the PreADME online application, the penetration through the BBB was predicted for the synthesized compounds and in general, they exerted moderate (0.1–2.0, C_brain_/C_blood_) to good (more than 2, C_brain_/C_blood_) permeation properties (Table 1). Cinnamamide **8f** exhibited the highest C_brain_/C_blood_ value (4.64) among the series of tested compounds, and it is worth noting that this ratio decreases as the chemical transformation on this core is performed, passing through *N*-(2-bromobenzyl) cinnamamide **7f** (3.39), 3,4-dihydroquinolin-2(1*H*)-one **6f** (2.07), and pyridophenanthridin-6-one **4f** (1.83) (Table 1 and Figure 3). In contrast, the C_brain_/C_blood_ ratio increased from cinnamamide **8c** (0.67), substituted with the methoxy group, to pyridophenanthridin-6-one **4c** (3.64), being the most promising nitrogen-containing tetracyclic molecule that could penetrate the BBB and exhibit a promising action within the CNS (Table 1).

Another important parameter in drug development is the oral bioavailability of potentially active compounds, and in this context, the small intestine is the major absorption target for oral drugs, since poor intestinal absorption will induce failure in the early stage of drug discovery studies [64]. To investigate transport mechanisms and to predict oral absorption, the Caco-2 cell monolayer model has been used to assess the in vitro human intestinal permeability of a drug, due to its morphological and functional similarity with human enterocytes, as well as a portion of the metabolic enzymes that are expressed in the human intestinal epithelium [65].

However, performing high-throughput screenings with this model is difficult due to its long culturing period (21 days) [47]. Thus, in silico models have emerged as ideal alternatives with a high degree of correlation between predicted and experimental values. According to the PreADMET tools, the prediction of Caco-2 cell permeability can be assessed, considering the same pH conditions (7.4) in which in vitro assays are performed. In this model, the oral permeability coefficient (P) is expressed in nm/s and compounds are categorized as low permeability agents (P < 4), middle permeability molecules (P = 4–70), and high permeability derivatives (P > 70). After assessing the intestinal absorption parameter using the Caco2 cell permeability model for all the synthesized compounds, all of them can be classified as middle permeability agents where the series of *N*-(2-bromobenzyl) cinnamamides **7** exhibited higher P_Caco-2_ values (Table 1).

The pharmacokinetic profile of the synthesized compounds would be completed with the prediction of the skin permeability coefficient (Kp), a parameter related to the molecular size and lipophilicity of each derivative. This assessment was performed using the online SwissADME database where the skin permeability is determined by the log*Kp* value; the more negative the value of logKp (with Kp in cm/s) is, the less skin permeant will be the molecule. In general, all the compounds exhibited moderate to good skin permeability properties where the series of compounds derived from cinnamamide **8f**, substituted with the ethyl group (**7f**, **6f**, and **4f**), resulted in the most promising agents among the series of tested compounds (Table 1).

The analysis of the predicted gastrointestinal absorption and brain access can be more easily achieved using the Brain or IntestinaL EstimateD permeation graph (BOILED-Egg), an accurate predictive model that combines the lipophilicity (LogP) and polarity (TPSA) of the tested small molecules [66]. The developed graphical representation (Figure 4), achieved with the SwissADME database, for the tested molecules in this study revealed that all the compounds, **4** and **6**–**7**, are located in the elliptical region where well-absorbed molecules can be found (Figure 4).

Examining it in detail, we observed that compounds of the series **7b** and **7f** were found in the white region and thus will passively be absorbed by the gastrointestinal tract, while the rest of derivatives **4**, **6**, **7,** and **8** will passively permeate the BBB as they are located in the yellow region. Considering that P-glycoprotein, a member of transmembrane transport proteins, acts as a barrier to intestinal drug absorption, for compounds located in the yellow region, *N*-aryl cinnamamides **8a**–**f** were the molecules that were predicted not to be effluated from the CNS by the P-glycoprotein, in contrast to molecules **4**, **6** and **7a**–**c** (Figure 4).

Finally, the toxicity profile of the prepared compounds, obtained with the PreADME application, revealed that all of them were mutagenic in the Ames assay, except for the series of *N*-(2-bromobenzyl)cinnamamides **7a**–**e** and **7f**. In addition, most of them exhibited positive carcinogenic activities except for pyridophenanthridin-6-ones **4a**–**f** (Table 1).

### 2.3. In Vivo Toxicity Assessment in Zebrafish Embryos

The Zebrafish (*Danio rerio*) has been established as a modern experimental vertebrate model for assessing the toxicity of diverse synthetic and natural small molecules (SMs) [67,68]. After the Organization for Economic Cooperation and Development (OECD) formalized the guideline for testing chemicals on fish embryos, under the Test Guideline 236: Fish Embryo Acute Toxicity test (FET) [69], Ali and co-workers described and standardized the toxicological assessment of SMs using zebrafish embryos in what is currently known as the ZFET test [70]. Thus, these previously reported methodologies were adapted in our laboratory with slight modifications for the toxicological assessment of the selected pyridophenanthridin-6-one **4f** and its corresponding precursors **6f**–**8f** (Table 2) [47].

These molecules were particularly selected to study their toxicological profile since they exhibited physicochemical properties and ADMET profiles among the series of all the synthesized compounds (Table 1). The toxicity of cinnamamide **8f** and its derivatives **6f**–**7f** and **4f** was compared with the toxicity of the selected positive controls camptothecin **13**, imiquimod **14**, orlistat **15**, ribavirin **16**, propranolol **17**, butylated hydroxyanisole (BHA) **18**, and eugenol **19**, a broad range of bioactive compounds with anti-cancer, antiviral, antioxidant, enzyme-inhibitor, and anesthetic activities (Table 2).

In general, the toxicity of the selected compounds increased as the chemical transformations were performed to transform cinnamamide **8f** (LC_50_ = 221.2 µM) into the corresponding pyridophenanthridin-6-one **4f** (LC_50_ = 67.81 µM). Although the amide **8f** was the less toxic compound, the substitution of the amide nitrogen by the benzylic group considerably increased the toxicity of *N*-(2-bromobenzyl) cinnamamide **7f** (LC_50_ = 136.91 µM), while the 3,4-dihydroquinolin-2(1*H*)-one **6f** exhibited moderate toxicity (LC_50_ = 151.20 µM), indicating that cyclic conformation was less toxic than the respective acyclic amide. Finally, the planarity, the conjugated system, and the smaller number of rotatable bonds in the pyridophenanthridinone core of **4f** can explain the high toxicity of this derivative (Table 2).

Regarding the selected positive controls **13**–**16**, camptothecin **13** was the most lethal compound, with an LC_50_ of 3.8 nM, while imiquimod **14** was the less toxic agent among this series of molecules, with an LC_50_ of 350.8 µM (Table 2). With the toxicity data of the selected reference compounds, a toxicity scale was established to describe the toxicological profile of the selected pyridophenanthridin-6-one **4f** and its corresponding precursors **6f**–**8f** (Figure 5).

## 3. Materials and Methods

### 3.1. General Procedures

Unless otherwise noted, all reactions were carried out with distilled and dried solvents. All work-up and purification procedures were carried out with reagent-grade solvents (purchased from Sigma-Aldrich and Merck, St. Louis, MO, USA) in the air. Thin-layer chromatography (TLC) was performed using Merck silica gel 60 F254 precoated plates (0.25 mm). Column chromatography was performed on Biotage^®^ Automated Liquid Chromatography System Isolera One^®^ using Biotage^®^ SNAP Ultra 25 um HP-Sphere 10 g silica gel cartridges (Uppsala, Sweden).

Infrared (FT-IR) spectra were recorded on a Lumex Infralum FT-02 spectrometer, and the wave numbers of the absorption peaks are listed in cm^−1^. Peaks/Bands are characterized according to the functional group. ^1^H NMR spectra were recorded on a Bruker Avance-400 (400 MHz) spectrometer. Chemical shifts are reported in ppm with the solvent resonance as the internal standard (CDCl_3_: δ 7.26 ppm; DMSO-*d*_6_: δ 2.50 ppm). Data were reported as follows: chemical shift, multiplicity (s = singlet, d = doublet, t = triplet, dd = doublet of doublets, br = broad, m = multiplet), coupling constants (Hz), and integration. ^13^C NMR spectra were recorded on a Bruker Avance-400 (400 MHz) spectrometer with complete proton decoupling. Chemical shifts are reported in ppm from solvent resonance as the internal standard (CDCl_3_: δ 77.00 ppm; DMSO-*d*_6_: δ 40.45 ppm). On DEPT-135 spectra, the signals of CH_3_ and CH carbons are shown with a positive phase (+), while CH_2_ carbons are shown with a negative phase (−). Quaternary carbons are not shown.

High-Resolution Mass Spectra (HRMS) were measured on a Thermo Fisher Scientific LTQ Orbitrap XL apparatus (Waltham, MA, USA). Melting points were measured on a Fisher Johns melting point apparatus and are uncorrected.

### 3.2. General Procedure for the Synthesis of N-Aryl Cinnamamides **8a**–**f**

In a 10 mL crimper vial equipped with a magnetic stir was added triethylamine (2 mmol) to a mixture of cinnamic acid **10** (2 mmol) and TBTU (2-(1*H*-benzotriazol-1-yl)-1,1,3,3-tetramethylaminium tetrafluoroborate) (2 mmol) dissolved in DMF (3.6 mL) in an ice bath. After 1 h, the respective aniline **11a**–**f** (2 mmol) was added dissolved in 3.6 mL of DMF, and the reaction was subjected to microwave heating for 10 min at 100 °C. After cooling to room temperature, the crude mixture was extracted with ethyl acetate (3 × 10 mL) and the organic layer was washed with brine (1 × 10 mL) and dried over anhydrous Na_2_SO_4_. The solvent was removed under reduced pressure and the crude product was purified by silica-gel column chromatography (20–50% AcOEt in petroleum ether) to furnish the desired *N*-aryl cinnamamides **8a**–**f**.

### 3.3. General Procedure for the Synthesis of N-(2-Bromobenzyl)-N-Phenyl Cinnamamides **7a**–**f**

In a round-bottom flask (25 mL), the corresponding *N*-aryl cinnamamide **8a**–**f** (1.5 mmol) was dissolved in tetrahydrofuran (6 mL), then potassium *t*-butoxide (3 mmol) and 2-bromobenzyl bromide **12** (3 mmol) was added into the solution and the mixture was stirred at 70 °C for 4 h. The reaction was monitored through TLC. After the reaction was completed, the crude was extracted with dichloromethane (3 × 10 mL), the organic layer was washed with brine (1 × 10 mL) and dried over anhydrous sodium Na_2_SO_4_. The solvent was removed under reduced pressure and the crude product was purified by silica-gel column chromatography (10–40% AcOEt in petroleum ether) to furnish the desired *N*-(2-bromobenzyl)-*N-*phenyl cinnamamides **7a**–**f**.

### 3.4. General Procedure for the Synthesis of 4-Phenyl-3,4-Dihydroquinolin-2(1H)-Ones **6a**–**f**

In a 10 mL microwave reactor, the respective *N*-(2-bromobenzyl)-*N*-phenyl cinnamamide **7a**–**f** (1 mmol) was dissolved in 4 mL of trifluoroacetic acid (TFA) and the reaction was subjected to microwave heating for 40 min at 140 °C. After cooling to room temperature, the mixture was quenched with NaHCO_3_ (1 M), extracted with ethyl acetate (3 × 20 mL) and the organic layers were combined, dried over with Na_2_SO_4_, filtered, and concentrated under reduced pressure. The crude material was purified by silica gel column chromatography (30–60% AcOEt in petroleum ether) to furnish the desired 4-phenyl-3,4-dihydroquinolin-2(1*H*)-ones **6a**–**f**.

### 3.5. General Procedure for the Synthesis of Pyrido[3,2,1-de]Phenanthridin-6-Ones **4a**–**f**

A 5 mL microwave reactor was charged with 4-phenyl-3,4-dihydroquinolin-2(1*H*)-one **6a**–**f** (0.9 mmol), K_2_CO_3_ (3 equiv.), PivOH (0.3 equiv) as an additive, P(Cy)_3_ (0.1 equiv) as a ligand, and PdCl_2_(MeCN)_2_ as a catalyst. The vial was sealed and purged three times with argon and 3 mL of degassed *N,N*-dimethylacetamide (DMA) was added. The reactor was set for microwave-assisted heating for 1 h at 150 °C. Once the reaction finished, the system was cooled to room temperature and the crude mixture was filtered through a small pad of Celite, washing with ethyl acetate (3 × 20 mL). The organic layers were combined, dried over with Na_2_SO_4_, filtered, and concentrated under reduced pressure. The crude material was purified by silica gel column chromatography (40–60% AcOEt in petroleum ether) to furnish the desired pyrido[3,2,1-*de*]phenanthridin-6-ones **4a**–**f**.

### 3.6. In Silico and Bioinformatic Studies

In silico analysis for the calculation of the molecular descriptors and toxicity prediction, as well as the calculation of the drug-likeness, were obtained with the aid of an online SwissADME database. The information about the methods used to evaluate each property is available on the program website [http://www.swissadme.ch/index.php, accessed on 1 April 2022]. The toxicity risks, the affinity of the selected toxic targets, and the ADME properties were predicted with the online application PreADME, free available online at [https://preadmet.bmdrc.kr/adme-prediction/, accessed on 4 July 2022].

### 3.7. Toxicity Assessment of the Pyridophenanthridin-6-One **4f** and Its Corresponding Precursors **6f**–**8f** Using the Zebrafish Embryo Model

Wild-type adult zebrafish of both sexes were separated in two tanks (30 L each), according to their gender, at 26 ± 2 °C under natural light-dark photoperiods. Fishes were fed twice daily, and the water quality was recorded weekly, in order to acclimate the fishes for at least two weeks before experiments begin. For the reproduction of the adult fishes, small breeding tanks were set up in the evening previous to the experiment, each containing three males and one female specimen. The tanks were isolated until the next morning when the lights switched on and the natural mating occurred, without any perturbation.

The adult fishes were returned to their corresponding tank and the embryos were collected, pooled, and washed with E3 medium and transferred into a 92 mm glass Petri dish. Further, the dead, delayed, malformed, and unfertilized embryos were identified under a dissecting microscope and removed by select aspiration with a pipette. This last procedure was repeated at 12 and 20 hpf in order to remove the unfit embryos. Throughout this period of time, the embryos were kept at 28 ± 2 °C in an incubator under natural light–dark photoperiods. The selected embryos of 24 hpf from the Petri dish were gently distributed into 96-well plates, placing a single embryo and 200 μL of E3 medium per well.

Adult zebrafish were cared for and used according to the Guide of the National Institute of Health for Care and Use of Laboratory Animals, keeping them healthy and free of any signs of disease. The Ethics and Research Committee of the Heart Institute of Bucaramanga approved the protocol under the Acta Number 050 on 26 May 2019.

### 3.8. Determination of Zebrafish Embryo LC_50_

For this experiment, in total 72 embryos were required per sample in order to run three independent experiments in three different plates, and each compound was evaluated three times in the same plate, allowing the evaluation of four samples per plate. In the range of concentrations established by a geometric series, starting from 12.5 and finishing at 1250 μM, the determination of the LC_50_ (expressed in μmol of compound/L of solution) was based on the cumulative mortality after 72 h of chemical exposure (96 hpf). Each embryo was examined under a dissecting microscope and the statistical analysis was made using Regression Probit analysis with SPSS for windows version 19.0. Data are expressed as the standard error of the mean (SEM) of three different experiments in triplicate.

## 4. Conclusions

In conclusion, a novel series of pyrido[3,2,1-*de*]phenanthridin-6-ones were synthesized and evaluated in in silico and in vivo models. Based on a linear strategy, the nitrogen-containing tetracyclic system was efficiently prepared from cinnamamide intermediates through a sequential intramolecular Friedel-Crafts alkylation and a direct C–H arylation as key steps.

Due to the novelty of this system, and recognizing its biological and pharmaceutical potential, an in silico study was performed to predict the physicochemical properties, as Lipinski’s rule of five, of the new series of pyrido[3,2,1-*de*]phenanthridin-6-ones and their respective intermediates, identifying the drug-likeness of compound **4f** and its precursors **6f**–**8f**. In addition, ADMET predictions were also performed to explain the pharmacokinetics of the synthesized compounds. Finally, we described the toxicity of these leading compounds using the zebrafish embryo model, establishing the toxicological profile of pyrido[3,2,1-*de*]phenanthridin-6-one **4f** and determining how this profile changed from cinnamamide **8f** through the chemical transformations that derived on the target compounds. Our work will be a good starting point for further investigations related to the chemistry and biology of these interesting natural product-like nitrogen-containing tetracyclic molecules.

## Data Availability

Not applicable.

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
