# Peer review of "Synthesis, In Silico and In Vivo Toxicity Assessment of Functionalized Pyridophenanthridinones via Sequential MW-Assisted Intramolecular Friedel-Crafts Alkylation and Direct C–H Arylation"

_molecules, 2022, doi:10.3390/molecules27238112_

Round 1
Reviewer 1 Report
It is excellent work. The chemical approach is very good, as well as the synthesis protocol and the characterization of the compounds.
- The authors have developed a new method to obtain pyrido[3,2,1-de]phenanthridine-6-ones. The method is new, fast and effective.
- This synthetic method is new, and efficient, that could be applied to obtain other related molecules.
- The synthesis method is new. The application in organic chemistry will be of great relevance.
- Perhaps the ADMET study part of the article could be simplified.
- Conclusions consistent with the topic of the article, and arguments presented.
- References are appropriate.
Author Response
Please, se a file

Reviewer 2 Report
I consider that this article is well presented and written. Then, it can be accepted as is.
Author Response
Please, see a file

Reviewer 3 Report
In the paper entitled “Synthesis, in silico and in vivo toxicity assessment of functionalized pyridophenanthridinones via sequential MW-assisted 3 intramolecular Friedel-Crafts alkylation and direct C–H arylation, the authors reported synthesis of novel pyrido[3,2,1-de]phenanthridin-6-ones using cinnamamide intermediates. The toxicological profile of the final compounds and intermediates, was explored through in silico computational methods. The acute toxicity toward zebrafish embryos was also determined in the study.
Comments:
1. The authors should highlight the importance of pyridophenanthridinones as pharmacologically active scaffolds and should elaborate on its significance as biologically active molecule.
2. Mechanism of reaction needs to be illustrated and explained in more detail in the manuscript with the help of figures.
3. The compounds are well characterized.
Author Response
Please, see a file